# Low and Ultra-Low HER2 in Human Breast Cancer: An Effort to Define New Neoplastic Subtypes

**DOI:** 10.3390/ijms241612795

**Published:** 2023-08-14

**Authors:** Mariausilia Franchina, Cristina Pizzimenti, Vincenzo Fiorentino, Maurizio Martini, Giuseppina Rosaria Rita Ricciardi, Nicola Silvestris, Antonio Ieni, Giovanni Tuccari

**Affiliations:** 1Department of Human Pathology in Adult and Developmental Age “Gaetano Barresi”, Section of Pathology, University of Messina, 98125 Messina, Italy; mariausilia.franchina@studenti.unime.it (M.F.); vincenzo.fiorentino@unime.it (V.F.); maurizio.martini@unime.it (M.M.); nicola.silvetris@unime.it (N.S.); aieni@unime.it (A.I.); 2Department of Biomedical, Dental, Morphological and Functional Imaging Sciences, University of Messina, 98125 Messina, Italy; cristina.pizzimenti@unime.it; 3Department of Medical Oncology, Papardo Hospital, 98125 Messina, Italy; calaienco@hotmail.com

**Keywords:** HER2 expression, breast cancer, neoadjuvant treatment, HER2-low carcinomas, HER2-ultra-low carcinomas

## Abstract

HER2-low and ultra-low breast cancer (BC) have been recently proposed as new subcategories of HER2 BC, supporting a re-consideration of immunohistochemical negative scores of 0, 1+ and the 2+/in situ hybridization (ISH) negative phenotype. In the present review, we outline the criteria needed to exactly distinguish HER2-low and ultra-low BC. Recent clinical trials have demonstrated significant clinical benefits of novel HER2 directing antibody–drug conjugates (ADCs) in treating these groups of tumors. In particular, trastuzumab-deruxtecan (T-Dxd), a HER2-directing ADC, has been recently approved by the US Food and Drug Administration as the first targeted therapy to treat HER2-low BC. Furthermore, ongoing trials, such as the DESTINY-Breast06 trial, are currently evaluating ADCs in patients with HER2-ultra low BC. Finally, we hope that new guidelines may help to codify HER2-low and ultra-low BC, increasing our knowledge of tumor biology and improving a targetable new therapeutical treatment.

## 1. Introduction

Breast cancer (BC) is the most frequently diagnosed cancer in women worldwide, and it continues to be a relevant cause of death [1], even though new strategies in clinical–therapeutic management have been introduced, improving patients’ outcomes in terms of overall survival (OS), progression-free survival (PFS), and the pathology complete response (pCR) [2]. At first, therapeutic strategies for BC were guided by the expression of hormonal receptors (HR) and the Ki67 proliferation index [3]. Successively, the rising importance of the prognostic role of the anti-human epidermal growth factor receptor 2 (anti-HER2) brought about the establishment of a new classification of BC and the development of new therapeutic strategies based on the molecular characteristics of each tumor [3,4,5]. This classification includes four categories of BCs with different prognostic and predictive values [4,5]. In detail, Luminal A category is constituted by HER2-negative tumors that express HR at different levels [4,5]. Moreover, Luminal B tumors can be ER-positive, HER2-negative, or ER and HER2-positive with low levels of progesterone (PR) and may present different genetic mutations, resulting in less differentiated and more aggressive tumors compared to Luminal A tumors [6]. The third category is represented by triple-negative BC (TNBC) tumors, which do not express HR nor HER2 and are characterized by more aggressive behavior, a younger age of incidence, the worst prognosis, fewer treatment options, and are often associated with the BRCA1 mutation [7]. Finally, HER2-positive tumors are characterized by the overexpression of HER2 due to HER2 gene amplification, aggressive behavior, and poor prognosis [4]. In this way, the assessment of HER2 status by using immunochemistry (IHC) and in situ hybridization (ISH) represents a mandatory parameter for the correct characterization of and therapeutic choice for BC patients. Recently, the American Society of Clinical Oncology and the College of American Pathologists (ASCO/CAP) reaffirmed the HER2 testing guidelines to reduce confusion over terminology for reporting [8]. This came from the results of the 2022 DESTINY-Breast04 trial, which prompted the United States Food and Drug Administration (FDA) to expand the approval of the HER2 antibody–drug conjugate, trastuzumab-deruxtecan, from metastatic breast cancer patients with HER2 protein overexpression/amplification to also include metastatic patients with HER2 IHC 1+ or 2+/ISH negative results. Generally, in the past, HER2 IHC results have been subdivided in two categories: HER2-negative and HER2-positive, assigning a specific score to each sample based on the intensity and the pattern of expression in the cell membrane as well as the percentage of positive cells [9]. The HER2-negative category includes tumors with scores of 0, 1+, and 2+ without HER2 gene amplification during ISH, whereas the HER2-positive category includes tumors with scores of 3+ and 2+ with HER2 gene amplification during ISH [8]. The recent update suggests a new indication for trastuzumab-deruxtecan when HER2 is not overexpressed or amplified, although the immunohistochemical score is 1+ or 2+ without amplification by in situ hybridization. However, data about BC classed as IHC 0 are still limited, and therefore, no complete evidence has been reported regarding the different behaviors of these cancers, which probably do not respond to newer ADCs. Although current data do not support a new IHC 0 versus 1+ prognostic or predictive threshold for the response to trastuzumab-deruxtecan, this threshold is now relevant because of the trial entry criteria that supported its new regulatory approval. Consequently, it has become clinically relevant to distinguish IHC 0 from 1+, even if it appears premature to create new result categories of HER2 expression (HER2-low and HER2-ultra-low) [8].

The demonstration of BC cases with HER2 amplification have raised the question about the development of specific monoclonal antibodies against HER2 to realize a specific target therapy [10]. In this way, specific therapeutic agents, such as trastuzumab [11,12], pertuzumab [13,14], and lapatinib [15,16,17], have been developed, representing an extremely important tool in the precision era medicine that is able to significantly improve the life expectancy of HER2-positive BC patients [18]. In fact, HER2 overexpression is present in approximately 20% of patients with BC [19], and it is usually associated with more aggressive behavior [20], a higher risk of disease recurrence, and a shorter OS [20,21,22] in comparison to HER2-negative (HER2-) BCs. To date, no effective target treatments have been approved for HER2-negative BC patients, since several studies have demonstrated a beneficial effect of the addition of trastuzumab to chemotherapy only in tumors with HER2 overexpression [23,24,25]. However, recently, the introduction of novel anti-HER2 antibody–drug conjugates (ADC) strategies showed promising response rates and progression-free survival (PFS) in the so-called HER2-“low” BCs, a new definition attributed to tumors which present an HER2 IHC score of 1+ or 2+ without HER2 amplification [26,27,28]. Furthermore, evidence has been raised concerning the benefit of the treatment response in HER2-“ultra-low” BCs, characterized by an HER2 IHC score of 0 [29]. In detail, ADCs are modern versions of antibodies that target cell surface proteins, which are characterized by three components, such as an antibody directed to a cell surface protein, a cytotoxic agent, and a linker used to attach the cytotoxic agent to the antibody [26,27,28]. DM1 is the warhead of trastuzumab-emtansine (T-DM1), an inhibitor of microtubule polymerization that represented the first ADC approved for the treatment of solid tumors, while topoisomerase I inhibitors include deruxtecan (DXd), which is able to bind to and stabilize the topoisomerase I–DNA complex, inhibiting the relegation of DNA breaks and hampering DNA; the latter represents a warhead that is used in the recently approved anti-HER2 ADC trastuzumab-deruxtecan [28]. It has been demonstrated that the ADC trastuzumab-emtansine (T-DM1) presents poorer benefits for HER2-low breast cancer, while the novel ADC trastuzumab-deruxtecan (TDXd) showed significant improvements in both progression-free and overall survival in the results of the DESTINY-Breast04 trial [29].

This guideline update does not support the use of a HER2-low interpretive category, because the DESTINY-Breast04 trial did not include patients with HER2 IHC 0 results. In fact, there is no evidence to support the idea that IHC 1+ or 2+/ISH negative results are predictive of the trastuzumab-deruxtecan treatment response when compared to IHC 0 results. Therefore, the FDA expansion of approval to this group was only based on clinical trial eligibly criteria rather than a new predictive indication for HER2 testing.

In the present review, we focus on current applicative aspects concerning the BC management of patients with the new codified HER2-low and ultra-low statuses.

## 2. How to Currently Establish the HER2 Status

The first problem when defining HER2 overexpression in BC patients is considering how to measure the presence of HER2 in breast cancer cells. Although HER2 gene amplification has been correctly identified by fluorescence in situ hybridization (FISH) and in situ hybridization (ISH) [30], the immunohistochemical demonstration of the corresponding HER2 oncoprotein expression is considered to be the most practical approach in laboratories [31] (Table 1).

The ASCO/CAP guidelines for HER2 interpretation were first introduced in 2007 [32] stating that the HER2 status should be initially assessed by IHC and subsequently confirmed by FISH in cases of equivocal results [33]. The 2007 guidelines set a cutoff value of 30% positive neoplastic cells for HER2 IHC, and the HER2 gene result is amplified if the HER2/chromosome enumeration probe 17 (CEP17) ratio is >2.2 or the HER2 copy number is >6.0 in the FISH analysis [33]. In 2013, the revised ASCO/CAP guidelines [34] focused on further decreasing false negative results that could affect the recruitment of patients who could effectively benefit from a targeted therapy. A positive score (3+) was defined by the presence of complete, intense, and circumferential membrane staining in >10% of tumor cells, while an equivocal score (2+) was defined by the presence of incomplete and/or a weak to moderate circumferential membrane in >10% of the invasive tumor cells or the presence of intense, complete, and circumferential membrane staining in ≤10% of the invasive tumor cells. Otherwise, negative scores (1+ and 0) were defined by the presence of incomplete and weak membrane staining in >10% of the invasive tumor cells or the absence of staining or barely perceptible staining in <10% of tumor cells, respectively.

FISH analysis has been sometimes considered as the first choice to determine the HER2 status (Figure 1A,B), but more frequently, it has been applied to verify an equivocal score (2+) as an amplified HER2 gene [34]. BC cases are classified as amplified when HER2/CEP17 ratio is ≥2.0 or the HER2/CEP17 ratio is <2.0 with average HER2 copy number of ≥6.0 [34]. Regarding the ISH analysis, it is largely applied in cases of equivocal IHC results (score 2+). The 2018 ASCO/CAP guidelines take in consideration five different ISH patterns [8]. HER2 is amplified in group 1 (HER2/CEP17 ratio ≥ 2.0 and average HER2 copy number ≥ 4.0), not amplified in group 5 (HER2/CEP17 ratio < 2.0 with average HER2 copy number < 4.0), and equivocal in group 2 (HER2/CEP17 ratio ≥ 2.0 and average HER2 copy number < 4.0), group 3 (HER2/CEP17 ratio < 2.0 and average HER2 copy number ≥ 6.0), and group 4 (HER2/CEP17 ratio < 2.0 and average HER2 copy number ≥ 4.0 and <6.0) [8] (Table 2). For these latter three groups, it is crucial to include a concomitant rigorous IHC check concerning all embedded neoplastic material to achieve the most accurate evaluation of the HER2 status. In any case, repeated ISH tests should performed with the observer blinded to the previous results recounting the ISH (Table 1). If the results remain as before, in groups 2 and 4, the result will be HER2-negative. Only for group 3, the average HER2 copy number >/= 6 will be HER2-positive [35,36]. Obviously, the ISH rejection criteria have to be applied in a rigorous fashion in all BC cases (Table 2).

In clinical practice, these guidelines have been used to discriminate HER2-negative BCs from HER2-positive BCs, the latter being eligible for target therapy with anti-HER2 monoclonal antibodies, such as trastuzumab, pertuzumab, and margetuximab, which are able to improve the clinical outcomes of these tumors [23,24,25,35,36,37]. However, only 20% of BCs are HER2-positive, leaving the rest with fewer treatment options, confined to endocrine therapy and/or chemotherapy with less clinical benefits and more side effects [38,39,40].

Nevertheless, HER2 status interpretation, and subsequently, an accurate classification of BC patients, can be affected by several factors, with HER2 heterogeneity being the most important one, as documented elsewhere for other neoplasms [33,41,42]. In fact, intratumoral HER2 heterogeneity has been observed in various type of cancer, including a subset of BCs [43]. Moreover, several studies have shown that HER2 heterogeneity can affect the accurate HER2 status assessment, and it has been reported to be more common in tumors with HER2 equivocal status (2+) with an incidence of 16–36% HER2-positive BCs [44,45]. Different HER2 heterogeneous patterns can be found in BCs, as small clusters of amplified cells intermingle with clusters of non-amplified cells, or they can appear as distinguished fields of amplified and non-amplified cells [43,44]. Furthermore, HER2 heterogeneity can be found not only inside the primary tumor, but also between primary BCs and metastases or between the primary tumor and recurrent or metachronous lesions [33,46,47]. In the light of this evidence, some studies have reported that HER2 heterogeneity has been associated with a poor response to trastuzumab and poor clinical outcomes in HER2-positive primary and metastatic BCs [45,48,49]. In addition, HER2 heterogeneity has been associated with lower disease-free survival rates in HR-positive BCs as well as incomplete responses to neoadjuvant therapy [50]. Furthermore, several studies evaluating the concordance and/or discordance rates of HER2 IHC and ISH have shown a small percentage of cases (approximately 8–9%) [51], in which BCs were classified as HER2-negative IHC (score 0 and 1+), while being assigned as amplified by ISH [51,52,53,54,55,56]. It must be underlined that the reproducibility of HER2 tests might be also affected by several pre-analytical and analytical issues [57]. The main causes of discrepancy include technical issues, such as sample handling and fixation, interpretation bias, and intratumoral HER2 heterogeneity, complicating the identification of HER2-low expression in terms of both false positive and false negative results [51,58]. Moreover, the above-mentioned pre-analytical and analytical conditions may have major impacts on accurate identification of lower HER2 cases, not to mention the uncertainty of whether HER2 0 BC will respond to T-DXd.

## 3. Identification and Definition of HER2-Low BC

Within the large cohort of HER2-negative BC there is significant heterogeneity with distinct biological features, especially in tumors with IHC scores of 1+ or 2+ but no amplification of ISH and in tumors with an IHC score of 0 [59,60]. Tumors classified as HER2- IHC 1+/2+, ISH non-amplified have been recently defined as “HER-2 low-masked” or simply “HER2 low-positive”, representing about 45–55% of HER2-negative BC cases [26,29,59,60]. This new proposed subtype, although not official, has been considered “equivocal”, since it has some characteristics that are similar to HER2+ BC, but it actually does not appear to be targetable with standard anti-HER2 drugs [12]. Consequently, in a specific trial, no clinical benefit was achieved by adding only trastuzumab to adjuvant chemotherapy in high-risk invasive HER2-low BC [12]. Therefore, the conventional targeted HER2 therapy by monoclonal antibodies appears not to be an appropriate choice for HER2-low BC, since its activity basically consists of blocking the aberrant HER2 signaling (via dimerization inhibition) and the antibody-dependent cellular cytotoxicity [61]. However, the real paradigm shift in the HER2-low BC field is related to the use of new anti-HER2 antibody conjugated drugs (ADCs), whose good results in progression-free survival (PFS) and the response rate (RR) have been shown in several studies [62,63,64] and are deeply demonstrated by the results of the DESTINY-Breast 03 trial, which documents the significant benefit in terms of prognosis determined only by the novel ADC trastuzumab T-DXd [29]. This evidence, summarized at the ESMO 2021 congress [65], also revealed significant benefits of new ADC drugs in patients with metastatic HER2-low BC [66,67]. Consequently, in August 2022, the US Food and Drug Administration approved trastuzumab T-DXd for the treatment of patients with HER2-low metastatic BC, adding this new treatment to the National Comprehensive Cancer Network Guidelines for this subgroup of patients [66,67,68,69].

From a molecular perspective, HER2-low BCs are associated with several mutations, such as PIK3CA (31%), GATA3 (18%), TP53 (17%), and ERBB2 (8%), with a higher prevalence of FGFR1 amplification (defined as ≥10 copy number gain) [70]. Moreover, the HR status seems to be associated with HER2-low expression in BC, being more common in tumors with HR expression (65%) rather than triple-negative BCs (TNBC) (37%) [62].

The evaluation of the HER2-low status has not been formally defined, as no specific procedures have yet been established [26]. The details about the score concern either the weak to moderate complete membrane staining observed in more than >10 of tumor cells (IHC 2 + equivocal not amplified) or the incomplete membrane staining that is faint/barely perceptible in more than 10% of tumor cells (IHC 1+). Unfortunately, the assessment of the HER2-low status using conventional testing techniques may present elements inaccurately and, therefore, reproducibility evaluations of HER2 testing in HER2-low cases may reveal temporal as well as interobserver heterogeneity [2]. To help to standardize protocols, the US FDA recently approved the VENTANA PATHWAY anti-HER2/neu (4B5) rabbit monoclonal primary antibody for the IHC assessment of HER2 expression as the first companion diagnostic test; therefore, it was tested and used in the DESTINY 04 trial in metastatic BS patients to identify and select only HER2-low cases [71]. The Ventana system will remain the first but probably not the only one approved to identify HER2-low patients, as new evidence has shown that the Dako system detects HER2 expression with higher sensitivity compared to the Ventana system, not only in ISH-positive BC tumors, but also HER2 tumors without gene amplification (IHC 1+/2+), thus adding more patients identified as HER2-low patients and selected for new ADC targeted therapies [72]. In light of these suggestions, it is now mandatory to better standardize protocols to establish which test is more appropriate for stratifying HER2-negative as well as identifying HER2-low cases [68]. As a consequence, to accurately define HER2-low cases, implement validation and quality control systems may be applied in a reproducible format, recruiting more additional patients who may potentially be eligible for HER2-targeted therapies [68,73]. Artificial intelligence (AI) and machine learning programs have been shown to perform well in terms of speed and accuracy in assessing the HER2 status and predicting the anti-HER2 treatment response, but specific pathologist training in reading IHC-ISH test results remains the current gold standard [74,75,76].

On the other hand, the HER2-low status should not be confused with the intratumor heterogeneity phenomenon [77]; the latter event may be generally responsible for a different positive response to targeted therapy in tissue microarrays (TMAs) of BC [77]. Recently, several studies have tried to find the presence of a predictive marker that is strictly connected with HER2-low patients that would take advantage of the use of newer ADC- T-DxD therapy [69,78,79,80]. In order to increase the number of HER2-low BC patients suitable for inclusion in HER2-targeted therapy [73], several studies have started to classify all new possible HER2 subgroups by analyzing the clinical and molecular landscapes of HER2-low BC more accurately, where possible. Therefore, a new classification has been proposed to apply a more specific ADC therapy for each subcategory, even in cases of HER2-low expression [18,29,81].

In this way, many efforts have been performed by clinical trials to evaluate the real efficacy of HER2 target-therapy in low-HER2 expression patients, focalizing data concerning mismatch repair (MMR) genes/proteins, and other related biomarkers [82,83,84,85,86]. This evidence is based on very sporadic findings, since HER2 activating mutations are described in less than 2% of BC cases with a higher frequency in HR-positive BCs in comparison to TNBC and likewise in lobular carcinomas than in ductal ones [83].

## 4. Identification and Definition of HER2-Ultra-Low BC

Although HER2-0 scored BCs are generally considered to inadequately respond to monoclonal antibodies, it has been reported that, among these patients, there is a cohort to be identified, recently defined as HER2-ultra-low [87]. This proposed subtype could represent another subcategory that is eligible for ADC targeted therapies. It is immunohistochemically characterized by faint/barely perceptible and incomplete staining in <10% of tumor cells without amplification on FISH [87]. Ongoing studies, such as the DESTINY-Breast06 trial, are currently evaluating ADCs in patients with HER2-ultra-low BC, who could benefit from new conjugated HER2-targeted therapies in the cohort previously scored as 0 HER2 [61]. However, this minimal HER2-ultra-low expression could probably be sufficient for the novel ADCs to exert their specific cytotoxic effect on the neoplasm. In this “ultra-low” phenotype, activating genetic mutations not related to the IHC status have been reported, representing an alternative way of activating the HER2 pathway in BC [84]. In detail, the V777L ERBB2 mutation is an activating mutation, as it strongly increases the phosphorylation of signaling proteins, indicating enhanced activity of the tyrosine kinase [84]; for this reason, BC cases with neoplastic cell HER2 V777L-mutated can be administrated with TK inhibitors (like lapatinib and neratinib) [84]. Furthermore, MutL deficiency (connected with mismatch repair system alteration) represents a mutation related to endocrine treatment resistance, which appears in 15–17% of estrogen-receptor positive (ER+)/HER2-negative BC cases [85,88,89]. It may be hypothesized that the loss of MutL expression could activate HER2 without receptor overexpression; consequently, MutL loss has been proposed as a marker to stratify ER+/HER2-negative BC patients who are probably responsive to anti-HER agents [89]. So, HR+/HER2-negative BC with a molecular mutation with MutL loss show a good response to a combination of anti-HER2 drugs and endocrine treatments [86,90]. Finally, despite molecular characteristics, the very slight HER2 expression shown in the HER2-ultra-low subgroup could also be due to the limitations of pathological testing, in which false negative results could be artefactually related to an inadequate formalin fixation process or insufficient sensitivity of the IHC assay [91,92].

## 5. New Treatment Options for HER2-Low and Ultra-Low Patients

The development of novel antibody–drug conjugates (ADCs) has recently revolutionized the therapeutic scenario of HER2-low tumors, demonstrating a survival benefit in this setting (Table 3). Trastuzumab-deruxtecan (T-DXd) is a novel ADC consisting of a monoclonal antibody targeting HER2 and a potent payload, the topoisomerase I inhibitor deruxtecan. These components are linked together with a ratio of 1:8 by a tetrapeptide linker. After the internalization of T-DXd, deruxtecan is released by lyposomale enzymes and inhibits topoisomerase I which, in turn, leads to the breakage of double-stranded DNA [93]. Furthermore, deruxtecan is able to pass through the cell membrane and kill tumor cells nearby, regardless of HER2 expression. This is called the bystander effect [94]. This explains the increased activity of this agent in HER2-low tumors, as compared with older HER2-targeting agents, including previous-generation ADCs, such as T-DM1. In a phase I trial, T-DXd showed promising antitumor activity in 54 pretreated HER2-low metastatic breast cancer (MBC) patients with a reported 37% overall response rate (ORR) and a median PFS of 11.1 months [95]. These encouraging results were confirmed in a phase III trial, the DESTINY Breast 04 study. In this study, patients with HER2-low MBC who had received one or two prior lines of chemotherapy were randomized 2:1 to receive T-DXd or the physicians’ choice of chemotherapy. The trial demonstrated the superiority of T-DXd over the physicians’ choice of treatment in terms of both the PFS (9.9 vs. 5.1 months, respectively; HR 0.50; *p* < 0.001) and OS (23.4 vs. 16.8 months, respectively; HR 0.64; *p* = 0.001) [69]. A subgroup analysis by hormone receptor (HR) expression showed that the advantage was consistent in the HR+ group, with a median PFS of 10.1 months in the T-DXd cohort vs. 5.4 months in the physicians’ choice of treatment arm (HR 0.51, *p* < 0.001) and a median OS of 23.9 months vs. 17.5 months, respectively (HR 0.64, *p* = 0.003) [69].

Only a small number of HR-negative patients, i.e., triple-negative breast cancer (TNBC), were included in the trial (11.3% of all the enrolled patients). In an exploratory analysis of this subgroup HR−, a median PFS of 8.5 months in the T-DxD group vs. 2.9 months (HR 0.46) was shown, and a median OS of 18.2 months in the T-Dxd cohort vs. 8.3 months in the physician’s choice of treatment group was reported [69]. Due to these impressive data, T-DXd represents a novel important therapeutic option for HER2-low BC and a further treatment option for HER2 low TNBC.

An ongoing phase 3 study is evaluating the superiority of T-DXd vs. chemotherapy in patients with HR+ HER2 low MBC (DESTINY Breast 06, NCT04494425). In addition to single agent use, several studies are investigating the combination of novel HER2-targeting ADCs and immune checkpoint inhibitors (ICIs), based on a strong preclinical rationale. Preclinical studies have reported synergistic activity between T-DXd and ICIs targeting PD-1 and CTLA-4, supporting the clinical development of T-DXd–immunotherapy combinations [96,97]. Preliminary data from arm 6 of the phase 1/2 BEGONIA trial evaluating the combination T-DXd with the PD-L1 inhibitor durvalumab in firstline HR−/HER2-low MBC patients have been recently presented [98].

Durvalumab plus T-DXd showed promising early safety and efficacy results in firstline HER2-low-expressing TNBC with a 66.7% ORR, irrespective of the PD-L1 expression. T-DXd was also evaluated in combination with the PD-1 inhibitor nivolumab in pretreated MBC in a phase 1b trial (NCT03523572) [99]. The combination was associated with a safety profile consistent with the expected toxicities of each drug and activity (ORR 65.6% in HER2+ and 50% in HER2-low with a median PFS of 11.6 months and 7.0 months, respectively) in line with the single agent T-DXd, questioning the additional benefit provided by the combinatorial use. Potential reasons for the differential activity seen with T-DXd plus immunotherapy combinations between firstline and pretreated patients might be attributable to a changed tumor immune microenvironment after multiple lines of treatment.

Other ADCs in development with the same target but a distinct payload targeting HER2-low expression are trastuzumab duocarmazine (SYD 985) and disitamab vedotin (RC48-ADC). Trastuzumab duocarmazine (SYD 985) is a new ADC consisting of the monoclonal antibody trastuzumab, which is covalently linked via a linker to a DNA-alkylating agent, duocarmazine. In a phase 1 trial that included 49 patients (32 HR+ and 17 HR−) with HER2-low MBC, trastuzumab duocarmazine was shown to have promising antitumor activity. Interestingly, in the HR+ and HR− cohorts, ORR values of 28% and 40% were reported, respectively, with a median of PFS of 4.9 months for HR+ and a median of 4.1 months for HR− [64].

Disitamab vedotin is another novel ADC that combines a novel anti-HER2 antibody, hertuzumab, linked via a cleavable linker to a microtubule inhibitor. Preliminary data from a cohort of 48 patients showed promising tumor activity with an ORR of 40% and a median PFS of 5.7 months, with a greater benefit reported in HER2 2+/ISH negative patients (ORR 42.9%) vs. HER2 1+ patients (ORR 30.8%) [100].

## 6. Conclusions

Recently published data and ongoing clinical trials have defined HER2-low and ultra-low expression in breast cancer, suggesting that new categorization and a new standardized approach to HER2 evaluation in BC is fervently required. Accepting the new scoring system and the consequent subgrouping, the clinical treatment as well as patient management will be dramatically revolutionized. Therefore, the acquired knowledge on HER2-low and ultra-low BC has produced further efforts, including basic and translational research as well as clinical studies in this newly recognized targetable group of BC. The development of the novel ADCs has recently changed the therapeutic scenario of HER2 BC, demonstrating a survival benefit in these patients. Furthermore, in the precision medicine era, oncologists and pathologists will have the ability to better define HER2-low and ultra-low BC as separated categories that will be finally standardized by guidelines and eligible for new therapeutic options.

## Figures and Tables

**Figure 1 ijms-24-12795-f001:**
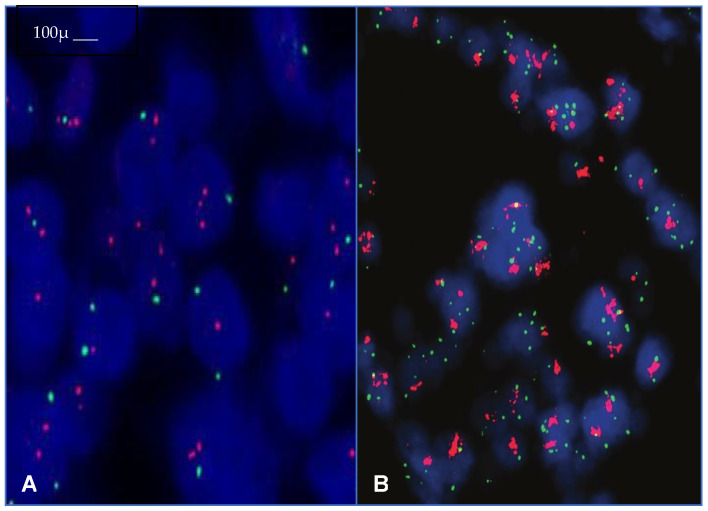
Peculiar features of the unamplified (**A**) or amplified (**B**) conventional FISH procedure in BCs (Original magnification, ×200, scale bar 100 μm).

**Table 1 ijms-24-12795-t001:** Interpretation of HER2 IHC staining and ISH analysis [8].

	HER2 Score 3+	HER2 Score 2+	HER2 Score 1+	HER2 Score 0
IHC	Complete, intense, and circumferential membrane staining in >10% of tumor cells	Incomplete and/or weak to moderate circumferential membrane staining in >10% of the tumor cells or the presence of intense, complete, and circumferential membrane staining in ≤10% of the tumor cells	Incomplete or faint/barely perceptible membrane staining in >10% of tumor cells	Incomplete or faint/barely perceptible membrane staining in ≤10% of tumor cells
				No staining
ISH	AMPLIFIED HER2/CEP17 ratio of ≥2.0 or HER2/CEP17 ratio < 2.0 with average HER2 copy number ≥ 6.0		NOT AMPLIFIED HER2/CEP17 ratio < 2.0 with an average HER2 copy number of <4.0. Notes: If the IHC result is 2+, recount ISH by having an additional observer, blinded to previous ISH results, count at least 20 cells that include the area of invasive cancer with IHC 2+ staining If reviewing the count by the additional observer changes the result into another ISH category, the result should be adjudicated as per internal procedures to define the final category. If the count remains an average of <4.0 HER2 signals/cell and an HER2/CEP17 ratio > 2.0, the diagnosis is HER2-negative with a comment.	
	HER2-POSITIVE		HER2-LOW	HER2-ULTRA-LOW
				HER2-NEGATIVE
Treatment	Trastuzumab-emtansine (T-DM1)		Trastuzumab-deruxtecan (T-DXd)	Other recommended regimens NCCN Guidelines^®^ Insights: Breast Cancer, Version 4.2023

**Table 2 ijms-24-12795-t002:** ISH Human Epidermal Growth Factor Receptor 2 Testing in Breast Cancer: ASCO-CAP Guideline Update.

ISH Interpretation	ISH Rejection Criteria
The entire ISH slide should be scanned prior to counting at least 20 cells or IHC should be used to define the areas of potential HER2 amplification. If there is a second population of contiguous cells with increased HER2 signals/cell and this cell population consists of >10% of tumor cells on the slide (defined by image analysis or visual estimation of the ISH or IHC slide), a separate count of at least 20 non-overlapping cells must also be performed within this cell population and reported.	Reject and repeat if: Controls are not as expected; The observer cannot find and count at least two areas of the invasive tumor; >25% of signals are unscorable due to weak signals; >10% of signals occur over the cytoplasm; The nuclear resolution is poor; The autofluorescence is strong. Report the HER2 test result as indeterminate as per the parameters described.

**Table 3 ijms-24-12795-t003:** Available data from current preclinical and clinical trials of novel ADCs.

Drug	Population	Clinical Trial	Result
Trastuzumab-Deruxtecan (T-DXd)	Pretreated HER2-low MBC	0 [95]	ORR = 37% PFS = 11.1 months
Trastuzumab-Deruxtecan (T-DXd)	HER2-low MBC pretreated with chemotherapy	NCT03734029 (DESTINY Breast 04—phase III) [69]	In HR+ patients: PFS = 9.9 months OS = 23.4 months In HR− patients: PFS = 8.5 months OS = 18.2 months
Trastuzumab-Deruxtecan (T-DXd)	HR+ HER2-low MBC	NCT04494425 (DESTINY Breast 06—phase III)	ongoing
T-DXd + anti-PD-L1	Preclinical study [96]		Enhanced antitumor effect by an increase in T-cell activity and upregulation of PD-L1 expression in xenograft mouse models
T-DXd + CTLA-4	Preclinical study [97]		Enhanced antitumor effect by increases in tumor-infiltrating CD4 and CD8
T-Dxd + Durvalumab	HER2-low locally advanced/metastatic TNBC	NCT03742102 (BEGONIA—phase Ib/II) [98]	ORR = 66.7% ongoing
T-Dxd + Nivolumab	Pretreated HER2-low MBC	NCT03523572 (phase Ib) [99]	ORR = 50% PFS = 7 months
Trastuzumab + duocarmazine	Pretreated HER2-low MBC	NCT02277717 (phase I) [64]	In HR+ patients: ORR = 28% PFS = 4.9 months In HR− patients: ORR = 40% PFS = 4.1 months
Hertuzumab + Disitamab Vedotin	Pretreated HER2-low MBC	NCT02881138 NCT03052634 (phase I/Ib) [100]	ORR = 40% PFS = 5.7 months

## Data Availability

Not applicable.

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
