# Peer review of "Low and Ultra-Low HER2 in Human Breast Cancer: An Effort to Define New Neoplastic Subtypes"

_ijms, 2023, doi:10.3390/ijms241612795_

Round 1
Reviewer 1 Report (New Reviewer)
This is a review article on the very actual topic of treating breast cancer showing HER2-low and ultra-low protein expression with new drugs. The review is correct in the meaning of summarizing the knowledge, the literture, the related trials and the upcoming problems in connection with determining the HER2 status of a tumor, however, it does not give any new elements that could be added to the subject as an original contribution. The text is well-written, the illustrations (tables) are helpful. The reference list is (almost) complete.
1. What does it add to the subject area compared with other published material?
- Nothing. It is review of the existing knowledge with no added new information. And this is normal in a review article. There are several published review articles on the same topic with exactly the same content. The question is whether a journal will or not publish a review on this actual topic.
2. *What specific improvements should the authors consider regarding the methodology?* What further controls should be considered?
- As it is a review article, there are no controls. The methodology is the usual one for a review article.
3. Are the conclusions consistent with the evidence and arguments presented and do they address the main question posed?
- Yes, they are.
4. Please include any additional comments on the tables and figures.
-The tables are perfect as their content is the same as in related guidelines and published articles.
Author Response
Dear reviewer,
thank you very much for your nice comments on our manuscript.
Reviewer 2 Report (New Reviewer)
This review summarizes the current knowledge on clinical behaviour, classification approaches, and novel ADC agents targeting breast cancer cells that exhibit low HER2 expression (Low and ultra-low HER2). The review points to the need of standardized classification schemes of breast cancer entities related to HER2 expression/amplification, presents current approaches to distinguish tumors expressing subtle amounts (ultra low) from truly HER2-negative ones, and underscores the strong clinical relevance of ADC-targeting HER2, as there is a chance of improving breast cancer management by combining standard chemo-endocrine treatment with ADC, which may influence disease outcome of many patients. Thus, the topic of the article is of general interest. The article is well structured by tables and figures.
1. Do you consider the topic original or relevant in the field? Does it address a specific gap in the field?
- The topic is original in the field, at least in terms of the unresolved question regarding classification of HER2 low breast cancers
2. What does it add to the subject area compared with other published material?
- It provides an overview of current publications in the field
3. What specific improvements should the authors consider regarding the methodology? What further controls should be considered?
- Not applicable. The manuscript must be spell/syntax checked
4. Are the conclusions consistent with the evidence and arguments presented and do they address the main question posed?
- Yes
5. Are the references appropriate?
- Yes
rigorous spell and syntax checks must be done throughout the manuscript. Some examples are given as follows:
L38: ER-2 should read ER
L40: syntax: …, resulting IN less differentiated…
L48-51: check syntax of sentence
L65: spell check: responding
L77: spell check: higher risk..
L95: syntax: It has been demonstrated THAT
L99-102: check syntax of sentence
L109: should read HER2
Figure 1 legend: replace exemplificative for another word
L176: …, while being assigned amplified by ISH
L183…, the word NOR does not fit
L195: use TARGETED therapy throughout the manuscript
Author Response
Dear Reviewer 2,
Thank you very much for your valuable suggestions and comments on our manuscript, as requested we have opportunately modified (underlined by yellow ink) our manuscript, in detail:
L38: ER-2 should read ER
We have corrected the sentence.
L40: syntax: …, resulting IN less differentiated…
We have corrected the sentence.
L48-51: check syntax of sentence
We have corrected the sentence, substituting it with appropriate one.
L65: spell check: responding
We have corrected the sentence.
L77: spell check: higher risk..
We have corrected the sentence.
L95: syntax: It has been demonstrated THAT
We have corrected the sentence.
L99-102: check syntax of sentence
We have corrected the sentence.
L109: should read HER2
We have corrected the sentence.
Figure 1 legend: replace exemplificative for another word
We have corrected the legend.
L176: …, while being assigned amplified by ISH
We have corrected the sentence.
L183…, the word NOR does not fit
We have corrected the sentence, substituting it with appropriate one.
L195: use TARGETED therapy throughout the manuscript
We have corrected the sentence.
Typographic errors have been corrected by an English Teacher.
This manuscript is a resubmission of an earlier submission. The following is a list of the peer review reports and author responses from that submission.
Round 1
Reviewer 1 Report
In this review article entitled “Low and ultra-low HER2 in human breast cancer: an effort to define new neoplastic subtypes”, Franchina et al have attempted to outline the needed criteria to distinguish these subgroups in the light of recent developments in the success of a new generation of antibody-drug conjugate trastuzumab deruxtecan (T-Dxd). There are however significant omissions and inaccurate statements that are unacceptable and most unexpected, given that the first as well are corresponding authors are pathologists.
Page 1. Line 38. “Luminal A and Luminal B categories are considered HER2-negative tumors….” This is incorrect because Luminal B subtypes could be ER-positive, HER2-negative or ER and HER2-positive tumors.
Page 2, line 50. “The American Society of Clinical Oncology/College of American Pathologist…..with its last update in 2018…..”. This again is incorrect, because the latest update on this topic is in fact 2023. It is a very recent update, precisely related to the topic of this review article. It is unacceptable to have missed this recent update completely with failure to make relevant comments on the most updated recommendations.
Page 2, line 68. “…..recently the introduction of novel anti-HER2 antibody-drug conjugates (ADC) strategies showed promising response rate and progression-free survival….” Here, as well as in page 5, line 162, the authors did not point out the distinction between the ADC trastuzumab emtansine (T-DM1) which gave poorer benefit in HER2-low breast cancer, and the novel ADC trastuzumab deruxtecan (T-DXd) which indeed showed significant improvement in both progression-free and overall survival from the results of the DESTINY-Breast04 trial.
Table 1. The information for interpretation of ISH analysis is too simplified and incomplete. A footnote at least should be included to mention ISH values in group 2, 3 and 4 cases.
Page 4, line 114-115. “…regarding groups 2 and 4, if HER2 IHC is 2+, then the ISH result is marked as not amplified…”. There should be mention that repeat ISH should be performed, with observer blinded to previous result recounting the ISH. If the results remain the same as before, in group 2 and 4, result will be HER2 negative. Only for group 3, average HER2 copy number>/= 6 will HER2 be positive.
Page 5, line 171. The percentage values for GATA23, TP53 and ERBB2, nor the percentage values with HR expression and of TNBCs (line174), are not that given in the quoted references.
Page 5, second last paragraph, lines 202-203 “unexpected positive response to target therapy with monoclonal antibody to HER2-negative BC” and 209-210 “a more specific targeted therapy for each subcategory either in HER2-low expression”. It is unclear what these sentences are referring to.
There is no mention of the importance of pre-analytic and analytic factors that are likely to have major impact on accurately identifying low-Her2 cases, nor the uncertainty of whether HER2 0 tumors will respond to T-Dxd.
Although the quality of English language is generally acceptable, the authors do not make it clear what they are refering to in page 5, second last paragraph, lines 202-203 “unexpected positive response to target therapy with monoclonal antibody to HER2-negative BC” and 209-210 “a more specific targeted therapy for each subcategory either in HER2-low expression”.
Author Response
Revised version
Manuscript n. ijms-2490536
MESSINA, 12th July 2023
Estimed Editorial Officer,
Dear Reviewers,
First of all, we wish to thank to reviewers for their useful and appreciable suggestions. In detail, we have tried to answer point by point for each question raised, changing the manuscript and underlining sections with yellow ink.
Reviewer 1:
Page 1. Line 38. “Luminal A and Luminal B categories are considered HER2-negative tumors….” This is incorrect because Luminal B subtypes could be ER-positive, HER2-negative or ER and HER2-positive tumors.
We have adequately modified the sentence in which a mistake in the definition of Luminal A and B categories has been made.
Page 2, line 50. “The American Society of Clinical Oncology/College of American Pathologist…..with its last update in 2018…..”. This again is incorrect, because the latest update on this topic is in fact 2023. It is a very recent update, precisely related to the topic of this review article. It is unacceptable to have missed this recent update completely with failure to make relevant comments on the most updated recommendations.
We have introduced the last 2023 update of ASCO/CAP with the actual recommendations.
Page 2, line 68. “…..recently the introduction of novel anti-HER2 antibody-drug conjugates (ADC) strategies showed promising response rate and progression-free survival….” Here, as well as in page 5, line 162, the authors did not point out the distinction between the ADC trastuzumab emtansine (T-DM1) which gave poorer benefit in HER2-low breast cancer, and the novel ADC trastuzumab deruxtecan (T-DXd) which indeed showed significant improvement in both progression-free and overall survival from the results of the DESTINY-Breast04 trial.
We have performed the distinction between ADC trastuzumab emtansine (T-DM1 and ADC trastuzumab deruxtecan (T-DXd either in line 68 either in line 162.
Table 1. The information for interpretation of ISH analysis is too simplified and incomplete. A footnote at least should be included to mention ISH values in group 2, 3 and 4 cases.
We have added the requested footnote inside the Table 1.
Page 4, line 114-115. “…regarding groups 2 and 4, if HER2 IHC is 2+, then the ISH result is marked as not amplified…”. There should be mention that repeat ISH should be performed, with observer blinded to previous result recounting the ISH. If the results remain the same as before, in group 2 and 4, result will be HER2 negative. Only for group 3, average HER2 copy number>/= 6 will HER2 be positive.
We have rephrased the sentence line 114-115, introducing also the new table 2 in which interpretation as well as rejection criteria for ISH have been reported.
Page 5, line 171. The percentage values for GATA23, TP53 and ERBB2, nor the percentage values with HR expression and of TNBCs (line174), are not that given in the quoted references.
In order to better define the percentage values for GATA23, TP53 and ERBB2 we have corrected previous references introducing new quoted and corrected ones.
Page 5, second last paragraph, lines 202-203 “unexpected positive response to target therapy with monoclonal antibody to HER2-negative BC” and 209-210 “a more specific targeted therapy for each subcategory either in HER2-low expression”. It is unclear what these sentences are referring to.
We have adequately rephrased sentences, better explaining lines 202-203 and 209-210.
There is no mention of the importance of pre-analytic and analytic factors that are likely to have major impact on accurately identifying low-Her2 cases, nor the uncertainty of whether HER2 0 tumors will respond to T-Dxd.
We have added a paragraph concerning the relevance of pre-analytic and analytic conditions able to impact on low-HER2 cases.
Comments on the Quality of English Language
Although the quality of English language is generally acceptable….
The manuscript has been revised by an English scientific teacher as reported in the Acknoledgement Section.
Reviewer 2 Report
Comments to the Authors
The Authors review Breast cancer subtypes focusing on patients with low and ultra-low HER2. They propose to further consider the HER2 status in those patients considering them for ADC treatments.
Overall, adding more figures (IHC, ISH,OS, PFS, ORR) would help the reader to understand the main idea of the authors.
Detailed comments:
Table1: Add the reference to the ASCO guidelines to the title, Also, add the recommended treatments by ASCO and FDAas an additional bottom row.
Additional figures with examples real pictures of ISH and ISH would help the reader to what ISH/ISH score is given.
Lines 106 – 116: Regarding this complicated ISH guidelines by ASCO. It would be a great improvement for the readability of this review to summarize the ISH analysis and recommendations in a table/figure.
Line 149-151: This sentence should be corrected (my corrections in bold)
Within the large cohort of HER2-negative BC there is a significant heterogeneity, with distinct biological features, especially in tumors with IHC score 1+ or 2+ but ISH not amplified, and in tumors with IHC score 057,58.
Line 156: correct this sentence: remove “it”
...quently, in a specific trial it no clinical benefit has been achieved by adding only ...
Line 247 fix this title what is “E”? and?
NEW TREATMENT OPTIONS IN HER2 LOW E ULTRA-LOW PATIENTS
Line 260: Explain abbreviation ORR = Overall response rate
Table 2 and elsewhere:
Add figures with Overall survival, progression-free survival, ORR and/or hazard ratios curves showing the results of ADC in the DESTINY-Breast 03/04/06 and other trials.
Line 318: fix this sentence
…will dramatically revolutionize (what?).
Quite a lot of typos and non-english word order in sentences. Needs to be edited by an English speaker
Author Response
Revised version
Manuscript n. ijms-2490536
MESSINA, 12th July 2023
Estimed Editorial Officer,
Dear Reviewers,
First of all, we wish to thank to reviewers for their useful and appreciable suggestions. In detail, we have tried to answer point by point for each question raised, changing the manuscript and underlining sections with yellow ink.
Reviewer 2
Table1: Add the reference to the ASCO guidelines to the title, Also, add the recommended treatments by ASCO and FDAas an additional bottom row.
We have added as requested the reference concerning the updated ASCO guidelines, introducing also the treatment by an additional bottom row.
Additional figures with examples real pictures of ISH and ISH would help the reader to what ISH/ISH score is given.
We have introduced two exemplificative pictures of unamplified or amplified FISH analysis in BC.
Lines 106 – 116: Regarding this complicated ISH guidelines by ASCO. It would be a great improvement for the readability of this review to summarize the ISH analysis and recommendations in a table/figure.
We have added a new table (Table 2) in order to improve the explanation about ISH guidelines.
Line 149-151: This sentence should be corrected (my corrections in bold)
Within the large cohort of HER2-negative BC there is a significant heterogeneity, with distinct biological features, especially in tumors with IHC score 1+ or 2+ but ISH not amplified, and in tumors with IHC score 057,58.
Line 156: correct this sentence: remove “it”
...quently, in a specific trial it no clinical benefit has been achieved by adding only ...
Line 247 fix this title what is “E”? and?
NEW TREATMENT OPTIONS IN HER2 LOW E ULTRA-LOW PATIENTS
Line 260: Explain abbreviation ORR = Overall response rate
Table 2 and elsewhere:
Concerning the abovementioned mistakes, We have adequately corrected all points, explaining also the abbreviations in the manuscript and tables.
Add figures with Overall survival, progression-free survival, ORR and/or hazard ratios curves showing the results of ADC in the DESTINY-Breast 03/04/06 and other trials.
In the present Table 3 we have reported all available percentage data present in the trials with reference to ORR, PFS, OS. The opportunity to introduce for each row the correspondent figures requires a formal approval by the scientists that have performed the clinical study.
Line 318: fix this sentence …will dramatically revolutionize (what?).
We have adequately rephrased sentence, better explaining the meaning line 318.
Comments on the Quality of English Language
Quite a lot of typos and non-english word order in sentences. Needs to be edited by an English speaker
Typos and inappropriate words have been modified since the manuscript has been revised by an English scientific teacher, as reported in the Acknoledgement Section.
Round 2
Reviewer 1 Report
Line 53-54 Besides re-affirming the 2018 ASCO-CAP recommendation for HER2 testing, more recommendations were mentioned in the discussion section in the most update 2023 ASCO/CAP recommendations. These points should have been expounded upon in this review, e.g. why is there no change to prior recommendation, the importance of a new HER2 testing reporting footnote and best practices for identification and reporting of IHC 0 vs IHC 1+ results. This latest update still has not been quoted as reference..
Line 59. Reference 9 seems inappropriate because it deals with gastro-oesophageal malignancies
Lines 73-76 and 217. The authors continue to lump “novel anti-HER2 antibody-drug conjugates (ADC)” as one entity which showed “promising response…”, and only add as comment at the end of the paragraph to make the distinction between the two ADC drugs. The differences between these two ADCs should be explained from the beginning, including that of their composition and mechanism of action, thus accounting for the differences in efficacy.
Line 127-128. “The last ASCO/CAP update takes into consideration five different ISH patterns” It should be explicit the ACSO/CAP guidelines of 2018, because the latest update is 2023.
First manuscript, Page 5, line 171. The percentage values for GATA23, TP53 and ERBB2, nor the percentage values with HR expression and of TNBCs (line174), are not that given in the quoted references. Revised manuscript lines 222-226 are supposed to have quoted new references. However, the references quoted remain exactly the same.
First manuscript Page 5, second last paragraph, lines 202-203 “unexpected positive response to target therapy with monoclonal antibody to HER2-negative BC” and 209-210 “a more specific targeted therapy for each subcategory either in HER2-low expression”. It is unclear what these sentences are referring to. This remains unclear in the revised manuscript lines 254-258. It is the use of the term monoclonal antibody which makes it unclear. In line 209, it is mentioned “Therefore, monoclonal antibodies appear to be a not appropriate choice in HER2-low BC, since their activity basically consists in blocking the aberrant HER2 signalling....” It is inferred the term “monoclonal antibodies” in this manuscript refers to classic anti-HER2 therapies that conventionally target HER2 signaling. Whilst this seems to be the case for the sentence in lines 254-255, the next sentence together with its quoted references appears to refer to new treatment options, namely the use of the ADC T-DXd instead, rather than monoclonal antibody therapy.
Typographic errors are present that need to be corrected
Author Response
Dear Editor Officer,
as requested by the new round of observations made by reviewer n. 1, we have opportunately modified (underlined by green ink) our manuscript and we enclose a point by point revision.
In detail:
Line 53-54 Besides re-affirming the 2018 ASCO-CAP recommendation for HER2 testing, more recommendations were mentioned in the discussion section in the most update 2023 ASCO/CAP recommendations. These points should have been expounded upon in this review, e.g. why is there no change to prior recommendation, the importance of a new HER2 testing reporting footnote and best practices for identification and reporting of IHC 0 vs IHC 1+ results. This latest update still has not been quoted as reference.
We have corrected the paragraph changing 2018 in 2023, also quoting the corresponding actual reference.
Line 59. Reference 9 seems inappropriate because it deals with gastro-oesophageal malignancies
We have omitted the inappropriate reference 9, substituting it with appropriate one.
Lines 73-76 and 217. The authors continue to lump “novel anti-HER2 antibody-drug conjugates (ADC)” as one entity which showed “promising response…”, and only add as comment at the end of the paragraph to make the distinction between the two ADC drugs. The differences between these two ADCs should be explained from the beginning, including that of their composition and mechanism of action, thus accounting for the differences in efficacy.
As requested, we have introduced a more detailed explanation about the two ADCs, including composition and mechanism of action, accounting thus for the differences in efficacy.
Line 127-128. “The last ASCO/CAP update takes into consideration five different ISH patterns” It should be explicit the ACSO/CAP guidelines of 2018, because the latest update is 2023.
We have corrected the sentences.
First manuscript, Page 5, line 171. The percentage values for GATA23, TP53 and ERBB2, nor the percentage values with HR expression and of TNBCs (line174), are not that given in the quoted references. Revised manuscript lines 222-226 are supposed to have quoted new references. However, the references quoted remain exactly the same.
We have substituted the uncorrect references, either for percentage values for GATA23, TP53 and ERBB2, either in relation to values with HR expression and TNBCs.
First manuscript Page 5, second last paragraph, lines 202-203 “unexpected positive response to target therapy with monoclonal antibody to HER2-negative BC” and 209-210 “a more specific targeted therapy for each subcategory either in HER2-low expression”. It is unclear what these sentences are referring to. This remains unclear in the revised manuscript lines 254-258. It is the use of the term monoclonal antibody which makes it unclear. In line 209, it is mentioned “Therefore, monoclonal antibodies appear to be a not appropriate choice in HER2-low BC, since their activity basically consists in blocking the aberrant HER2 signalling....” It is inferred the term “monoclonal antibodies” in this manuscript refers to classic anti-HER2 therapies that conventionally target HER2 signaling. Whilst this seems to be the case for the sentence in lines 254-255, the next sentence together with its quoted references appears to refer to new treatment options, namely the use of the ADC T-DXd instead, rather than monoclonal antibody therapy.
The first paragraph should be attributed to the phenomenon concerning heterogeneity, as documented also in the reference 77. Successively, in others manuscript lines the term monoclonal antibody has been clarified, specifying the adequate the therapy ADC-TDxD in low-HER2 BC.
Typographic errors have been corrected by an English Teacher.